# Molecular Liquids versus Ionic Liquids: The Interplay between Inter-Molecular and Intra-Molecular Hydrogen Bonding as Seen by Vaporisation Thermodynamics

**DOI:** 10.3390/molecules28020539

**Published:** 2023-01-05

**Authors:** Sergey P. Verevkin, Dzmitry H. Zaitsau, Ralf Ludwig

**Affiliations:** 1Department of Physical Chemistry, University of Rostock, Albert-Einstein-Str. 27, 18059 Rostock, Germany; 2Department of Physical Chemistry, Kazan Federal University, 420008 Kazan, Russia; 3Department LL&M, University of Rostock, Albert-Einstein-Str. 25, 18059 Rostock, Germany; 4Department of Technical Thermodynamics, University of Rostock, Albert-Einstein Str. 2, 18059 Rostock, Germany; 5Leibniz-Institut für Katalyse an der Universität Rostock e.V., Albert-Einstein-Str. 29a, 18059 Rostock, Germany

**Keywords:** ionic liquid, hydrogen bonding, enthalpy of vaporisation, structure–property relationship

## Abstract

In this study, we determined the enthalpies of vaporisation for a suitable set of molecular and ionic liquids using modern techniques for vapour pressure measurements, such as the quartz crystal microbalance, thermogravimetric analysis (TGA), and gas chromatographic methods. This enabled us to measure reasonable vapour pressures, avoiding the problem of the decomposition of the ionic liquids at high temperatures. The enthalpies of vaporisation could be further analysed by applying the well-known “group contribution” methods for molecular liquids and the “centerpiece” method for ionic liquids. This combined approach allowed for the dissection of the enthalpies of vaporisation into different types of molecular interaction, including hydrogen bonding and the dispersion interaction in the liquid phase, without knowing the existing species in both the liquid and gas phases.

## 1. Introduction

At room temperature, methane, ethane, propane, and butane are gases, but pentane and hexane are already very volatile liquids. Admittedly, the weak van der Waals attractions are the obvious reason these non-polar molecules are held together in the liquid phase. At room temperature, the first representative of the homologous series of alkanols, methanol, is already a volatile liquid. The inter-molecular hydrogen bonding is the obvious reason for such behaviour of these polar molecules. The energetics of inter-molecular interactions of molecules in the liquid phase (and thus their volatility) can be quantitatively determined using vaporisation thermodynamics. According to the textbook, a vaporisation enthalpy is the amount of energy required to disrupt the network of inter-molecular interactions in the liquid and to transfer one mole of molecules in the ideal gas, where interactions between molecules are assumed to be absent. Admittedly, the enthalpy of vaporisation is the “bulk” thermodynamic quantity that fails to distinguish between the energetics of van der Waals forces and the energetics of hydrogen bonding coexisting in typical *molecular liquids*. Nevertheless, the vaporisation enthalpies, $\Delta_{l}^{g}H_{m}^{o}$, are reliably measured by various experimental techniques [1], challenging the task of separating and understanding the energetics of molecular interactions using these thermodynamic data. Indeed, in a number of recent publications, we have developed a few thermodynamically based methods to quantify hydrogen bond strengths (inter- and intra-HB) for molecular [2,3,4] and ionic liquids [5,6,7,8,9,10,11,12,13,14,15]. In this work, we develop and apply these methods to a series of molecular liquids (MLs) containing a six-membered aliphatic ring (alkyl-piperidines) and a hydroxyl group as a special feature (see Figure 1 left); we also apply these methods to corresponding ionic liquids (ILs) that also contain a six-membered aromatic ring (alkyl-pyridinium) with the hydroxyl attached to the alkyl chain (see Figure 1 right).

The association of the hydroxyl-containing molecules (e.g., alcohols, diols, amino alcohols, etc.) in the liquid phase via the OH···O and OH···N hydrogen bond types (see Figure 2) is a well-established phenomenon [2,3,4,9].

The formation of both the inter- and intra-types of association is theoretically feasible, but in practice, most molecules (even in amino alcohols and diols) in the liquid are connected via the OH···O and OH···N inter-molecular hydrogen bonds, which are more thermodynamically favourable [2,3,4]. In contrast, the amino alcohol molecule, which has left the liquid phase and freely flies in the gas phase (or in vacuum) without adjacency, has only one option—to build the intra-molecular H-bond, which brings the molecule into the energetic lowest conformations. Depending on the molecular flexibility and temperature, the number of coexisting intra-H-bonded conformations can be significant, making the quantification of the intra-HB strength in the gas phase a difficult task [2,3,4]. Similar complications apply to the hydroxyl-containing ionic liquids. In the liquid phase, the formation of both inter- and intra- OH···O and OH···N association types is also feasible (see Figure 2), but the presence of the large and charged anion indicates the additional competition for H-bonding modes in ILs. The determination of ratios and relationships between the possible HB modes is a challenging task that could only be partially solved with the help of molecular dynamics simulations [9]. It should be mentioned that the definition of intra-molecular hydrogen bonding in ionic liquids has a certain peculiarity. When a hydrogen bond occurs within two functional groups of the same molecule, it is called an intra-molecular hydrogen bond; however, in the case of hydroxyl-containing ionic liquids, the formation of the non-covalent bond occurs between the OH group and the anion of the same molecule [9].

Furthermore, in the gas phase, the interpretation of the intra-HB by the free-flying IL molecule is thwarted with complications. Although we have only recently shown that the OH group in the hydroxyl-containing ILs is definitely linked to the anion, the coexistence of numerous intra-molecular H-bonded conformers with comparatively low energy makes it difficult to properly quantify the HB strength.

We hope that the apparent structural similarity of molecules chosen for this work (see Figure 1) could enable the development and understanding of the interplay between inter-molecular (*HB*_inter_) and intra-molecular (*HB*_intra_) hydrogen bonding in both MLs and the corresponding ILs.

A proven method for determining the strength of the *HB*_inter_ hydrogen bond in alkanols is to use the method of homomorphs [16]. The idea behind this is to compare the vaporisation enthalpies of alkanols (R–OH) with those of the homomorphic alkanes, which are obtained by replacing the OH group with a CH_3_ group (R–CH_3_). The vaporisation enthalpies of such homomorphic alkanes essentially model the van der Waals forces contributions to the vaporisation enthalpies of the alkanols (R–OH). The difference in the enthalpy of vaporisation between an alkanol (R–OH) and its corresponding homomorph (R–CH_3_)
(1)$${\mathrm{HB}}_{\mathrm{inter}}=\Delta_{l}^{g}H_{m}^{o}\left(R–{\mathrm{CH}}_{3})-\Delta_{l}^{g}H_{m}^{o}(R–\mathrm{OH}\right)$$
could be taken as a crude measure of the contribution to the enthalpy of vaporisation due to the inter-molecular hydrogen bonding in alkanols. The numerical values for $\Delta_{l}^{g}H_{m}^{o}$ of some alkanols and their homomorphs are collected in Table 1.

The *HB*_inter_ values calculated according to Equation (1) are shown in column 5. They are practically independent of the chain length and the general level of a hydrogen bond strength of −25 kJ·mol^−1^ agrees with the results of IR and NMR spectroscopic methods [18,19].

## 2. Molecular Liquids: Strength of the Inter-Molecular Hydrogen Bonding

### 2.1. Series of Hydroxy-Alkyl-Piperidines

The application of the homomorphic method is straightforward and simple. The vaporisation enthalpies of the hydroxy-alkyl-piperidines required for this analysis were derived from the temperature dependence of their vapour pressures (see details in the Appendix A), which were measured using the transpiration method (the primary experimental data are given in Appendix A). The final experimental $\Delta_{l}^{g}H_{m}^{o}$ (298 K) values relevant to the discussion are presented in Table 2, column 2.

The vaporisation enthalpies of the corresponding N-alkyl-piperidines required for comparison were taken from our preliminary work [21] and checked for internal consistency (see Appendix A); the experimental $\Delta_{l}^{g}H_{m}^{o}$ (298 K) values essential for the discussion are presented in Table 2, column 4. The *HB*_inter_ values calculated according to Equation (1) for hydroxy-alkyl-piperidines are listed in column 5. Similar to aliphatic alcohols, they are (within their experimental uncertainties) hardly dependent on chain length, but the general hydrogen bond strength level of −13 kJ·mol^−1^ is two times lower than in alkanols. This level is quite explainable as the general level of −9 kJ·mol^−1^ is specific for the association in pure amines (see Appendix A), and consequently, the strength of the inter-molecular hydrogen bonds in amino alcohols decreases to the approximate mean between alcohols and amines. Some peculiarities of the chain-length dependence and influence of alkyl substituents on the *HB*_inter_ in amino alcohols have been just recently reported [2]. Essential to this discussion is that the six-membered ring specific to the hydroxy-alkyl-piperidines does not impact the *HB*_inter_ strength in these types of amino alcohols. To show a comparative example, *HB*_inter_ strength for the open-alkyl-chain N,N-di-ethyl-2-aminoethanol can hardly be distinguished from those in the hydroxy-alkyl-piperidines of interest in this work (see Table 2). Other examples can be found elsewhere [2,20,22].

### 2.2. Series of Hydroxy-Alkylbenzenes

It is now clear that the aliphatic six-membered ring specific to the hydroxy-alkyl-piperidines plays no role in the association intensity in such MLs. Considering that in the next step we want to perform the comparison with corresponding ILs that also contain a six-membered but aromatic ring, it makes sense to understand if the aromatic ring can influence the association in the hydroxy-alkyl-benzenes, C_6_H_5_-(CH_2_)_n_-OH, and roughly mimic the intended ILs structures. The vaporisation enthalpies of phenyl-methanol and phenyl-ethanol required for this analysis were taken from our previous work [23,24]. For 3-phenyl-1-propanol and 4-phenyl-1-butanol, we collected vapour pressures at different temperatures that are available in the literature (see Appendix A). The vaporisation enthalpies of homomorphic alkyl-benzenes are well-known from the literature [25]. The experimental $\Delta_{l}^{g}H_{m}^{o}$(298 K) values that are relevant to the discussion are given in Table 3 under column 2 and column 4.

The *HB*_inter_ values calculated according to Equation (1) for hydroxy-alkyl-benzene and given in Table 3, column 5 make it clear that even the aromatic ring cannot significantly affect the inter-molecular hydrogen bonding network in this type of ML. The common level of hydrogen bond strength of −25 kJ·mol^−1^ remains more or less the same as for aliphatic alkanols. A slight decrease in the *HB*_inter_ in 2-phenylethanol is later explained as a consequence of the interplay of the inter- and intra-HB specific to this molecule, where the OH group is in very close proximity to the phenyl ring.

Thus, after *HB*_inter_ analysis in aliphatic and aromatic alkanols, as well as in aliphatic amino alcohols bearing the six-membered ring, we could expect the *HB*_inter_ level in the similarly shaped hydroxyl-alkyl-pyridinium bis(trifluoromethylsulfonyl)imides (Figure 1 right) to be roughly comparable to those found in amino alcohols, provided that the NTf_2_ anions do not participate in the association of these ILs. Will our expectation come true?

### 2.3. Ionic Liquids: Strength of the Inter-Molecular Hydrogen Bonding

Admittedly, the reliable measurement of the vapour pressure of extremely low-volatility ionic liquids is an extremely demanding task [5,6,7,8,9,10,11,12,13,14,15]. The reason for this is that the conventional techniques used for ML studies are not applicable as they are not sensitive enough to detect the vanishingly small ILs vapour pressures. Increasing the temperature enables reaching measurable vapour pressures, but at such high temperatures, the thermal stability of ILs approaches, and the limit and decomposition products aggravate the measurements. To overcome these difficulties, several methods have been developed in our laboratory. A brief description of the quartz crystal microbalance (QCM) [26], thermogravimetric analysis (TGA) [27], and gas chromatographic (GC) [28] methods is given in the Appendix A. The TGA and QCM methods were applied to derive vaporisation enthalpies for the [N-C_n_OH-Py][NTf_2_] series (see Table 4). Complementary results for ionic liquids containing the [N-C_n_OH-Py] cation, but connected to the dicyanoamide [DCA] and tetrafluoroborate [BF_4_] anions were derived using the TGA and GC techniques. The details of the experimental conditions and results are compiled in Table 4.

From the analysis of the data compilation given in Table 4, it is interesting to note that in the series [N-C_n_Py][NTf_2_] with n = 3, 4, an expected increase in vaporisation enthalpies of ≈4 kJ·mol^−1^ per CH_2_ group is observed. At the same time, in the series [N-C_n_OH-Py][NTf_2_] with n = 2, 3, and 4, this trend is less pronounced with increasing chain length, indicating a specific influence of the OH group placed on the alkyl chain in close proximity to the charged cation.

The summary of results relevant for the discussion in the frame of this work is given in Table 5, column 2. The vaporisation enthalpies of the corresponding homomorphic ionic liquids containing the N-alkyl-pyridinium-cation required for comparison are compiled in Table 4, column 4. The *HB*_inter_ values calculated according to Equation (1) for the [N-C_n_OH-Py][NTf_2_] ionic liquids are listed in Table 5, column 5.

It was overoptimistic to expect that the anion would not participate in the hydrogen-bonding network in ionic liquids. Indeed, the overall level of the inter-molecular hydrogen bond strength of ≈−7 kJ·mol^−1^ (see Table 5) in the [NTf_2_]-containing ILs is two times lower than in the corresponding ML of hydroxy-alkyl-piperidines (see Table 2). The significant decrease in the *HB*_inter_ values of the ILs compared with the MLs is clear evidence that a considerable part of the OH groups in the ionic liquid structure are excluded from the OH···O inter-molecular hydrogen bonding network and are most likely involved in the formation of the intra-molecular hydrogen bonding with the anion. The quantification of the individual inter- and intra-HB contributions to the vaporisation enthalpies goes beyond the experimental methods used in this work, but it is interesting to point out some general trends that are evident for “bulk” *HB*_inter_ values in Table 5 and Table 6.

First, the *HB*_inter_ chain-length dependence within the series [N-C_n_OH-Py][NTf_2_] is absent, indicating more favourable OH-[NTf_2_]-anion interactions. Second, the *HB*_inter_ strength is slightly anion-dependent. As can be seen in Table 6, for ionic liquids with the [N-C_3_OH-Py]-cation with [DCA] and [BF_4_], the OH···O interactions are at the level of disappearing (≈−2 kJ·mol^−1^, see Table 6, column 5). However, in the [OMs]-containing ILs, the OH–anion interactions have already overcome the ML-specific OH···O hydrogen bonding among hydroxyl groups, which is now practically absent in these ILs (≈+9 kJ·mol^−1^, see Table 6, column 5). Thus, it can finally be concluded that the OH groups connected to the [N-C_n_OH-Py]-cations favour the intra-molecular type of hydrogen bonding rather than the inter-molecular type OH···O expected from MLs.

How does one determine the strength of intra-molecular hydrogen bonds in MLs and ILs?

## 3. Thermodynamic Methods: Strength of the Intra-Molecular Hydrogen Bonding

The strength of inter-molecular hydrogen bonding is easily measurable by mixing two components bearing the specific groups, e.g., alcohol and amine [32]. In the case of intra-molecular hydrogen bonding, the desired interaction takes place within the same molecule, so direct measurements are out of the question. As a rule, the strength of the intra-molecular hydrogen bond (*HB*_intra_) is determined by comparing the H-bonded and non-H-bonded frequencies, shifts, conformers, etc. In terms of vaporisation thermodynamics, the main challenge is to find a suitable reference system that mimics the desired molecule without the intra-HB. In our earlier work, we proposed the reference system based on conventional group additivity [2,3]. In this work, we improve this concept of group additivity (GA) and propose an alternative method based on the well-balanced equations (WBE).

### 3.1. Basics of the Group Additivity Concept (“Centerpiece” Approach)

Group additivity methods have been successfully used to predict vaporisation enthalpies of molecular [33,34] and ionic liquids [29,35]. The idea behind conventional GA methods is to split the experimental vaporisation enthalpies of molecules into relatively small groups in order to obtain well-defined numerical contributions for them. The prediction then proceeds as a construction of a framework of the desired model molecule from the appropriate number and type of these contributions. A comprehensive system of group contributions (or increments) is developed and covers the main classes of molecular organic compounds [33,34]. A general transferability of these increments to ionic liquids has only recently been demonstrated [29]. However, using this conventional GA method for ionic liquids composed of large organic cations and large organic/inorganic anions is impractical due to the very limited amount of available experimental enthalpies of vaporisation required for the parameterisation of specific groups. To overcome this limitation, we developed a general approach for assessing vaporisation enthalpies based on a so-called “centerpiece” molecule [29].

The idea of the “centerpiece” approach is to select a potentially large “centerpiece” molecule that has a reliable vaporisation enthalpy and that can generally mimic the structure of the molecule of interest. Then, the necessary groups are attached to the “centerpiece”, resulting in the construction of the desired molecule. For example, we selected N-ethyl-piperidine for the series of hydroxy-alkyl-piperidines and obtained the “centerpiece” -CH_2_-(piperidine) (see Figure 3 left) suitable for calculations of the framework by cutting off the CH_3_ group. We selected [N-C_3_-Py][NTf_2_] for the series of hydroxy-alkyl-piperidines and obtained the “centerpiece” -CH_2_-(Py-NTf_2_) (see Figure 3 right) suitable for calculations of the framework by cutting off the CH_2_ and CH_3_ groups.

Such an approach significantly increases the reliability of the prediction, as the main energetic contribution is already stored in the “centerpiece” and the appended groups taken from the ML compounds are exactly parameterised [33,34]. The development of the groups and other “centerpieces” used in this work is shown in Appendix A, and their numerical values are given in Appendix A.

### 3.2. Intra-HB Strength from Group-Additivity

Although we developed the “centerpiece” approach, it is worth noting that predicting the enthalpies of vaporisation of hydroxy-alkyl-piperidines and hydroxy-alkyl-pyridinium [NTf_2_] ionic liquids was not the aim of this work, as we could measure them for both series (see Table 4). The goal of developing groups and “centerpieces” (see Appendix A) was to create a “non-hydrogen-bonded” reference system for estimating the intra-molecular hydrogen-bonding strength in hydroxy-alkyl-piperidines and hydroxy-alkyl-pyridinium [NTf_2_] ionic liquids. Indeed, the existence of the intra-molecular hydrogen bonding in the hydroxy-alkyl-piperidines is well-known [36]; however, as shown in Figure 2, the framework of 1-piperidine-propanol collected from “centerpiece”, -CH_2_-(piperidine) and two additional groups does not contain intra-molecular hydrogen bonding a priori, as it is composed of groups in which intra-molecular hydrogen bonding is originally absent. Therefore, the difference between the experimental vaporisation enthalpy of 1-piperidine-propanol and that of its framework, $\Delta_{l}^{g}H_{m}^{o}$(framework) (see Figure 4) directly provides a measure of the intra-molecular hydrogen-bonding strength, *HB*_GA_, in this molecule:
(2)$${\mathrm{HB}}_{\mathrm{GA}}=\Delta_{l}^{g}H_{m}^{o}\left(\exp)-\Delta_{l}^{g}H_{m}^{o}(\mathrm{framework}\right)$$

The development of the “non-hydrogen-bonded” framework for calculating the intra-molecular hydrogen-bonding strength, *HB*_GA_, in 1-(3-hydroxypropyl)pyridinium [NTf_2_] follows the same pattern as is shown in Figure 4.

The *HB*_GA_ values calculated for 1-piperidine-propanol and for 1-(2-hydroxyethyl)pyridinium [NTf_2_] according to Equation (2), as well as for other MLs and ILs of interest for this work, are given in Table 1, Table 2 and Table 3 and Table 5 and Table 6 for comparison. For the sake of completeness, the $\Delta_{l}^{g}H_{m}^{o}$(framework) values denoted by *FW* are also given in these tables. The application of the GA method is straightforward and easy and allows for the estimation of intra-molecular hydrogen bond strengths, particularly conveniently in homologous series. However, this method required some preliminary work to develop the system of groups and the selection of the appropriate “centerpiece”.

### 3.3. Intra-HB Strength from Well-Balanced Equations

Another alternative way to create the “non-hydrogen-bonded” reference system is shown in Figure 5 for piperidine-ethanol and 1-(2-hydroxyethyl)pyridinium [NTf_2_].

The idea comes from the quantum chemical calculations, where the enthalpies of formations are derived using isodesmic, homodesmic, well-balanced reactions, etc. [37]. The main criteria for choosing the reactants is that the structures (or bond types) on the left and right should be as similar as possible and the enthalpy of such a reaction should be as close to zero as possible. To redirect this idea to enthalpies of vaporisation, we should set up a well-balanced equation where the participants at the left and right side are “dressed” with the same functional groups, but the groups are distributed in such a way that there are no other specific interactions on the left and right sides and only one task-specific interaction governs the right side of the equation. This idea is illustrated in Figure 5 using piperidine-ethanol and 1-(2-hydroxyethyl)-pyridinium [NTf_2_] as examples. The task-specific interaction in piperidine ethanol is the intra-molecular hydrogen bond. To extract this interaction, we balanced this molecule with n-butane, n-propyl-piperidine, and propanol so that the stoichiometry is retained but the intra-HB is missing from the left part of the equation. Therefore, the difference in enthalpy between the right and left sides can be considered as a measure of the intra-molecular hydrogen bonding strength, *HB*_WBE_, in this molecule:
(3)$${\mathrm{HB}}_{\mathrm{WBE}}=[\Delta_{l}^{g}H_{m}^{o}(\mathrm{piperidine}-\mathrm{ethanol},\;\exp)+\Delta_{l}^{g}H_{m}^{o}(n-\mathrm{butane},\;\exp)]\;-\;\;[\Delta_{l}^{g}H_{m}^{o}\left(1-\mathrm{propyl}-\mathrm{piperidine},\;\exp)+\Delta_{l}^{g}H_{m}^{o}(1-\mathrm{propanol},\;\exp)\right]$$

The same logistic was applied for *HB*_WBE_ values calculations for 1-(2-hydroxyethyl)pyridinium [NTf_2_] shown in Figure 5, as well as for other MLs and ILs of interest in this work. These *HB*_WBE_ values are given in Table 1, Table 2 and Table 3 and Table 5 and Table 6 for analysis.

### 3.4. Intra-HB Strength in ML and IL Series

It is quite obvious that there are no intra-molecular hydrogen bonds in aliphatic alcohols, but just to confirm that Equation (2) correctly worked, we applied this equation to the aliphatic alcohol series with the completely negative result (see Table 1, column 7). It is known that the weak intra-molecular bond is observed for both phenyl-methanol (or benzyl alcohol) and phenyl-ethanol [36]. This observation is now numerically verified (see Table 3), along with the weak strength of the intra-HB for both phenyl-methanol (≈3 kJ·mol^−1^) and for phenyl-ethanol (≈5 kJ·mol^−1^) from the GA and WBE methods. It is interesting to note that a very subtle interplay of intra- and inter-HB can be revealed for phenyl-methanol and phenyl-ethanol as calculated by both methods. Taking into account the average inter-HB strength of −25 kJ·mol^−1^ in aliphatic alcohols (see Table 1), it can be seen from Table 3 that the increase in intra-HB strength resulted in a decrease in inter-HB, particularly pronounced in phenyl-ethanol. This observation makes it clear that a non-negligible part of intra-molecularly bound molecules is also incorporated into the mainly inter-molecular hydrogen-bonded network.

For the series of hydroxy-alkyl-piperidines, the chain-length dependence of intra-HB was found to be hardly distinguishable, and the overall level of strength of −13 kJ·mol^−1^ (see Table 2, columns 7 and 8) was estimated using both methods. Interestingly, this level is quite comparable to that of the inter-HB strength in this series (see Table 2, column 5). This observation indicates that qualitatively, a significant amount of the intra-molecularly bound molecules is involved in the association. However, the enthalpy of vaporisation as a thermodynamic tool is unable to quantify the ratio between the inter- and intra-HBs.

For the series of hydroxy-alkyl-pyridinium [NTf_2_] ionic liquids, the chain-length dependence of intra-HB is also absent, and the overall level of strength of ≈−20 kJ·mol^−1^ (see Table 5, column 7 and 8) was estimated using both methods. Moreover, the MD simulations performed in our previous work [9] showed that the OH···O hydrogen bond between the OH in the alkyl chain and the oxygen in the anion is definitely present in the gas phase ion pairs (see Figure 6).

Furthermore, quantum chemical calculations on these ion pairs showed that the favourable OH···O hydrogen bond energies of about ≈−20 kJ·mol^−1^ in the hydroxy-alkyl-pyridinium [NTf_2_] ionic liquids [9]. Such a perfect quantitative agreement between the empirical and quantum chemical assessment of the intra-molecular hydrogen bond strength supports the validity of the empirical models used in this work.

This level of intra-HB of ≈−20 kJ·mol^−1^ in the hydroxy-alkyl-pyridinium [NTf_2_] ionic liquids is considerably higher compared with that of the inter-HB strength of −7 kJ·mol^−1^ in this series (see Table 5, column 5). However, apparently a certain part of the IL molecules still determines the degree of association of the ionic liquids. This delicate interplay between the intra- and inter-HB is also evident for ILs with an anion other than [NTf_2_]: for 1-(3-hydroxypropyl)pyridinium [BF_4_] and 1-(3-hydroxypropyl)pyridinium [DCA], the intra-HB strength increased to ≈−24 kJ·mol^−1^ (see Table 6, column 6) and the strength of the inter-HB decreased to ≈−2 kJ·mol^−1^ (see Table 6, column 5). Finally, for the [N-C_n_OH-Py][OMs] series, the intra-HB strength increased independently of the chain-length up to ≈−35 kJ·mol^−1^ (see Table 6, column 6), but the level of inter-HB strength was ≈+9 kJ·mol^−1^ in this series. Consequently, in the [DCA]-, [BF_4_]-, and [OMs]-containing ILs, the molecules are practically not associated via OH groups, and all OH groups are bound to the anions in the liquid phase.

## 4. Conclusions

For understanding inter- and intra-molecular hydrogen bonding in molecular and ionic liquids, we measured the enthalpies of vaporisation by applying a large variety of thermodynamic methods. Because ionic liquids exhibit vanishingly small vapour pressures at low temperature and start to decompose at the required high temperature, we developed new techniques for measuring the vapour pressures and determining the enthalpies of vaporisation, which included the use of quartz crystal microbalance, thermogravimetric analysis (TGA), and gas chromatographic methods. Applying the well-known “group contribution” methods for molecular liquids and the “centerpiece” method for ionic liquids, we were able to dissect the enthalpies of vaporisation into different types of molecular and ionic interaction, such as hydrogen bonding and dispersion interactions. Considering molecular mimics of the IL cations and thus combining both approaches even allowed for the distinguishing between inter- and intra-molecular hydrogen bonding, which is shown here for the first time.

## Figures and Tables

**Figure 1 molecules-28-00539-f001:**
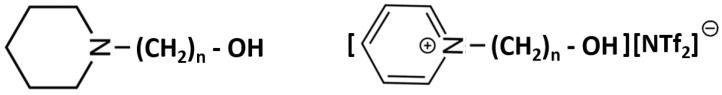
Structures of the hydroxyl-alkyl-piperidines (**left**) and hydroxyl-alkyl-pyridinium bis(trifluoromethylsulfonyl)imides = [N-C_n_OH-Py][NTf_2_] (**right**). The number of the methylene groups n = 2, 3, and 4.

**Figure 2 molecules-28-00539-f002:**
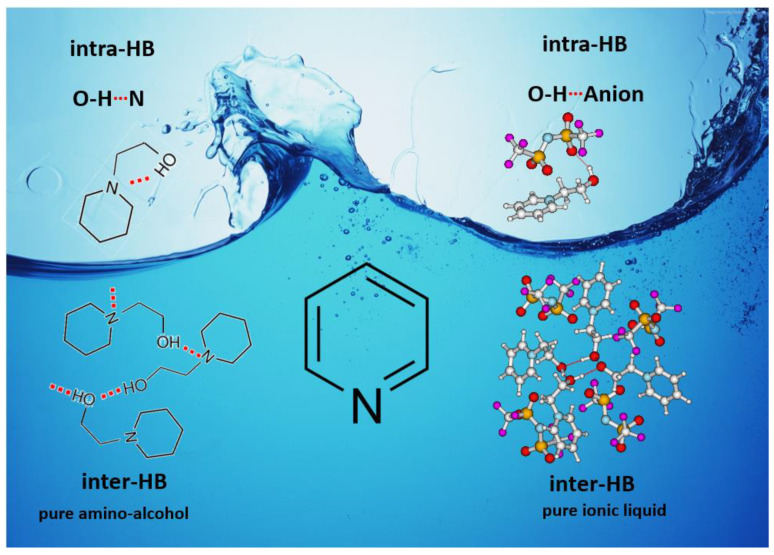
The inter-molecular (*HB*_inter_) and intra-molecular (*HB*_intra_) hydrogen bonding in MLs and corresponding ILs.

**Figure 3 molecules-28-00539-f003:**
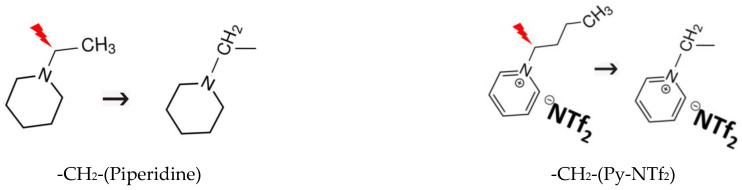
The development of the “centerpieces” for calculating the vaporisation enthalpies of hydroxy-alkyl-piperidines and hydroxy-alkyl-pyridinium [NTf_2_] ionic liquids.

**Figure 4 molecules-28-00539-f004:**
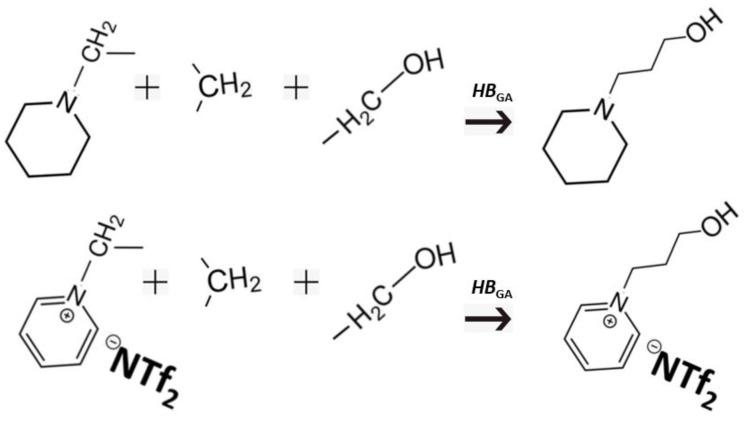
Group additivity: the development of the “non-hydrogen-bonded” frameworks for calculating the intra-molecular hydrogen-bonding strength, *HB*_GA_, in hydroxy-alkyl-piperidines and hydroxy-alkyl-pyridinium [NTf_2_] ionic liquids.

**Figure 5 molecules-28-00539-f005:**
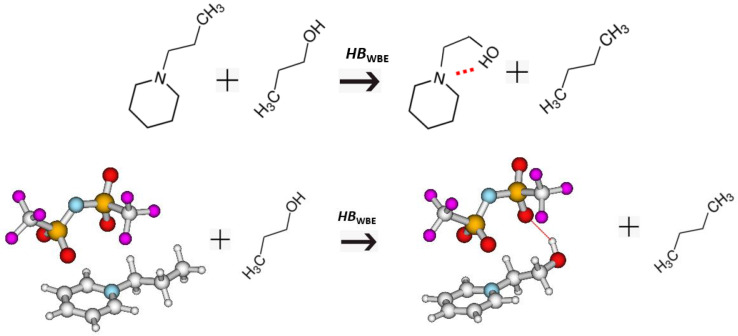
Well-balanced equations: the development of the “non-hydrogen-bonded” reference systems for calculating the intra-molecular hydrogen-bonding strength, *HB*_WBE_, in hydroxy-alkyl-piperidines and hydroxy-alkyl-pyridinium [NTf_2_] ionic liquids.

**Figure 6 molecules-28-00539-f006:**
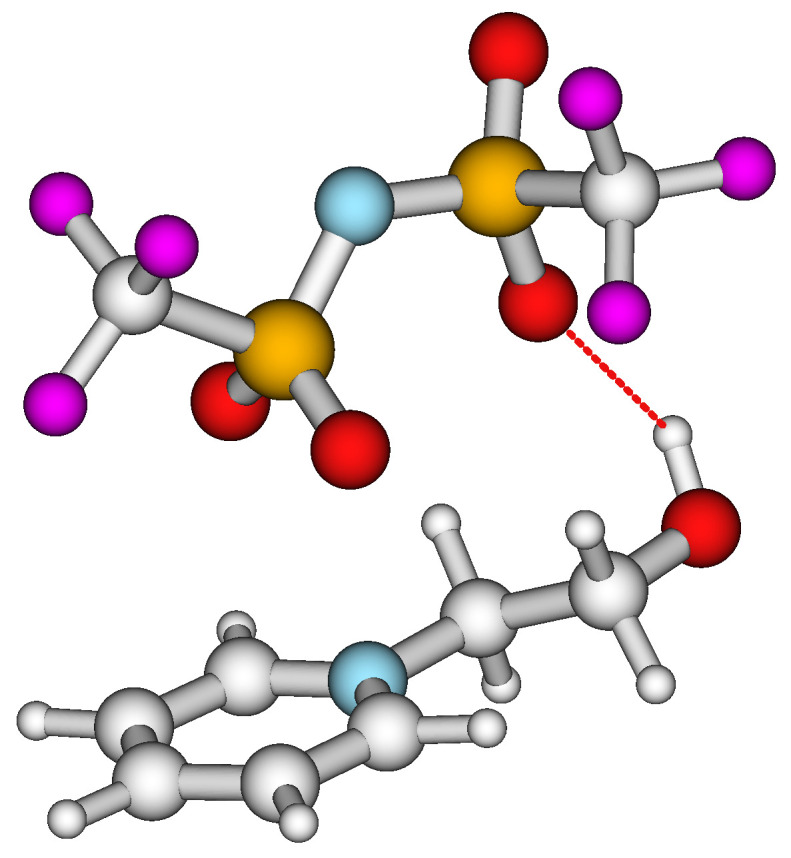
Gas phase ion pair of [C_2_OH-Py][NTf_2_] from snapshots of MD simulations. The existing hydrogen bond in the [C_2_OH-Py][NTf_2_] ion pair is shown as a dotted red line [9].

**Table 1 molecules-28-00539-t001:** Experimental enthalpies of vaporisation of alkanols and alkanes and the strength of inter- and intra-HB calculated from these values (at 298 K, in kJ·mol^−1^).

Compound	ΔlgHmo(exp) a	Compound	ΔlgHmo(exp) a	*HB*_inter_(Equation (1))	*FW* ^b^	*HB*_intra_(Equation (2))
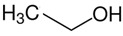 ethanol	42.5	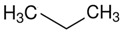 propane	16.3	−26.0	42.5	−0.2
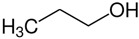 1-propanol	47.5	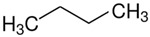 butane	22.4	−25.1	47.5	0.0
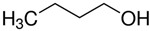 1-butanol	52.4	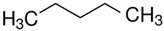 pentane	26.8	−25.6	52.5	−0.1
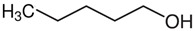 1-pentanol	57.0	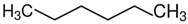 hexane	31.7	−25.3	57.5	−0.5

^a^ Experimental data from ref. [17] with uncertainties of ±0.2 kJ·mol^−1^. ^b^ Enthalpies of vaporisation referred to the frameworks (see text).

**Table 2 molecules-28-00539-t002:** Experimental enthalpies of vaporisation of hydroxy-alkyl-piperidines and N-alkyl-piperidines and the strength of inter- and intra-HBs calculated from these values (at 298 K, in kJ·mol^−1^).

Compound	$\Delta_{l}^{g}H_{m}^{o}\;(\exp){\;}^{a}$	Compound	$\Delta_{l}^{g}H_{m}^{o}\;(\exp){\;}^{b}$	*HB*_inter_ (Equation (1))	*FW* ^c^	*HB*_intra_ (Equation (2))	*HB*_intra_ (Equation (3))
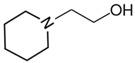 1-piperidine-ethanol	58.1 ± 0.4	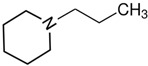 N-propyl-piperidine	44.9 ± 0.4	−13.2	72.0	−13.9	−11.9
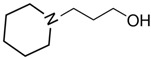 1-piperidine-propanol	62.1 ± 0.4	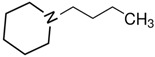 N-butyl-piperidine	48.9 ± 0.2	−13.2	77.0	−14.9	−12.4
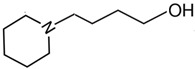 1-piperidine-butanol	70.0 ± 2.6	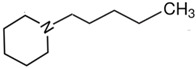 N-pentyl-piperidine	53.7 ± 1.0	−16.3	82.0	−12.0	−9.0
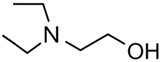 N,N-diEt-2-aminoethanol	52.6 ± 0.2 [20]	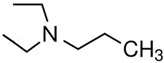 N,N-diEt-propylamine	39.8 ± 0.2 [17]	−12.8	66.2	−13.6	−12.3

^a^ Experimental data from Appendix A. ^b^ Experimental data from Appendix A. ^c^ Enthalpies of vaporisation referred to the frameworks (see text).

**Table 3 molecules-28-00539-t003:** Experimental enthalpies of vaporisation of hydroxy-alkyl-benzene and alkyl-benzenes and the strength of inter- and intra-HB calculated from these values (at 298 K, in kJ·mol^−1^).

Compound	$\Delta_{l}^{g}H_{m}^{o}\;(\exp)$	Compound	$\Delta_{l}^{g}H_{m}^{o}\;{(\exp)\;}^{a}$	*HB*_inter_ (Equation (1))	*FW* ^b^	*HB*_intra_ (Equation (2))	*HB*_intra_ (Equation (3))
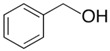 phenyl-methanol	65.8 ± 0.5 [23]	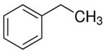 ethylbenzene	42.3	−23.5	68.5	−2.7	−2.7
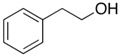 2-phenyl-ethanol	66.7 ± 0.3 [24]	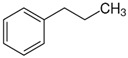 1-propylbenzene	46.2	−20.5	73.5	−6.8	−4.6
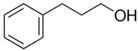 3-phenyl-1-propanol	75.1 ± 2.4 [Appendix A]	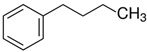 1-butylbenzene	50.8	−24.3	78.5	−3.4	−1.3
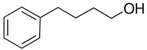 4-phenyl-1-butanol	80.2 ± 3.0 [Appendix A]	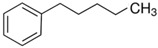 1-pentylbenzene	55.1	−25.1	73.5	−3.3	−0.2

^a^ Experimental data from ref. [16] with uncertainties of ±0.2 kJ·mol^−1^. ^b^ Enthalpies of vaporisation referred to the frameworks (see text).

**Table 4 molecules-28-00539-t004:** Experimental vaporisation enthalpies of pyridinium-based ionic liquids ^a^.

IL	Method	*T* _range_	*T* _av_	$\Delta_{l}^{g}H_{m}^{o}\left(T_{\mathrm{av}}\right){\;}^{b}$	$\Delta C_{p,m}^{o}{\;}^{c}$	$\Delta_{l}^{g}H_{m}^{o}\;\left(298\;K\right){\;}^{d}$
		*K*	*K*	kJ·mol^−1^	J·mol^−1^·K^−1^	kJ·mol^−1^
1	2	3	4	5	6	7
[N-C_2_OH-Py][NTf_2_]	TGA	532–592	562	125.2 ± 1.1	68	143.1 ± 3.8
	QCM	379–428	403.6	134.6 ± 1.0		141.8 ± 1.7
						**142.0 ± 1.6** ^e^
[N-C_3_OH-Py][NTf_2_]	QCM	389–436	412.5	136.8 ± 1.0	72	145.0 ± 1.9
	GC		298			143.6 ± 3.0 ^f^
						**144.6 ± 1.9** ^e^
[N-C_4_OH-Py][NTf_2_]	QCM	359–409	382.6	143.0 ± 0.8	76	149.4 ± 1.5
[N-C_3_OH-Py][DCA]	TGA	532–592	563.7	141.6 ± 2.0	81	163.1 ± 4.7
	GC	298				163.9 ± 3.0 ^g^
						**163.7 ± 2.5** ^e^
[N-C_3_OH-Py][BF_4_]	TGA	532–592	563.7	134.5 ± 2.1	67	152.3 ± 4.1
[N-C_3_H_7_-Py][NTf_2_]	QCM	375.1–422.4	398.2	127.9 ± 0.5	66	134.5 ± 1.4
[N-C_4_H_9_-Py][NTf_2_]	QCM	377.6–422.3	399.5	131.0 ± 0.5	70	138.0 ± 1.5
[N-C_2_OH-Py][OMs]	QCM	394.1–444.2	419.1	136.0 ± 1.0	74	145.0 ± 2.1
[N-C_3_OH-Py][OMs]	QCM	379.2–451.2	412.7	136.8 ± 1.0	81	146.1 ± 2.1

^a^ Uncertainties of vaporisation enthalpy ($\Delta_{l}^{g}H_{m}^{o}$) are twice the standard uncertainties. Abbreviations for anions: [DCA] = [NC-N-CN] = dicyanoamide; [OMs] = [CH_3_SO_3_] = methanesiulfonate. ^b^ Vaporisation enthalpy measured in the specified temperature range and in reference to the average temperature $T_{\mathrm{av}}$. ^c^ Heat capacities differences were taken from Appendix A. ^d^ Vaporisation enthalpy $\Delta_{l}^{g}H_{m}^{o}\left(T_{\mathrm{av}}\right)$ were treated using Equations (S2) and (S3) with the help of heat capacity differences from Appendix A to evaluate the enthalpy of vaporisation at 298 K. The final uncertainties of vaporisation enthalpy are expanded, taking into account the uncertainty of the heat capacity difference $\Delta_{l}^{g}C_{p,m}^{o}$, which was assigned to be ±20 J·K^−1^·mol^−1^. ^e^ Weighted mean value (uncertainty was taken as the weighing factor). ^f^ From Appendix A. ^g^ From Appendix A.

**Table 5 molecules-28-00539-t005:** Experimental enthalpies of vaporisation of hydroxyl-alkyl-pyridinium [NTf_2_] and alkyl-pyridinium [NTf_2_] and the strength of inter- and intra-HB calculated from these values (at 298 K, in kJ·mol^−1^).

Cation	$\Delta_{l}^{g}H_{m}^{o}\;(\exp){\;}^{a}$	Cation	$\Delta_{l}^{g}H_{m}^{o}\;(\exp){\;}^{a}$	*HB*_inter_ (Equation (1))	*FW* ^b^	*HB*_intra_ (Equation (2))	*HB*_intra_ (Equation (3))
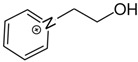 1-(2-hydroxyethyl)pyridinium	142.0 ± 1.6	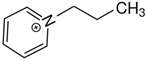 1-propyl-pyridinium	134.5 ± 1.4	−7.5	162.1	−20.1	−17.6
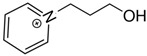 1-(3-hydroxypropyl)pyridinium	144.6 ± 1.9	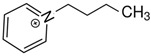 1-butyl-pyridinium	138.0 ± 1.5	−6.6	165.7	−21.1	−19.0
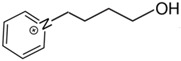 1-(4-hydroxybutyl)pyridinium	149.4 ± 1.5	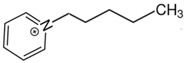 1-pentyl-pyridinium	141.7 ± 1.8 [29]	−7.7	169.3	−19.9	−17.6

^a^ Experimental data from Table 4. ^b^ Enthalpies of vaporisation referred to the frameworks (see text).

**Table 6 molecules-28-00539-t006:** Experimental enthalpies of vaporisation of hydroxyl-alkyl-pyridinium and alkyl-pyridinium ionic liquids containing [DCA], [BF_4_], and [OMs] anions and the strength of inter- and intra-HB calculated from these values (at 298 K, in kJ·mol^−1^).

IL	$\Delta_{l}^{g}H_{m}^{o}\;(\exp){\;}^{a}$	IL	$\Delta_{l}^{g}H_{m}^{o}\;(\exp)$	*HB*_inter_ (Equation (1))	*HB*_intra_ (Equation (3))
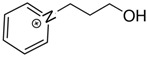 **DCA** 1-(3-hydroxypropyl)pyridinium	163.7 ± 2.5	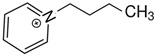 **DCA** 1-propyl-pyridinium	162.1 ± 4.1 ^b^	−1.6	−24.0
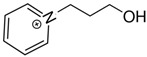 **BF_4_** 1-(3-hydroxypropyl)pyridinium	152.3 ± 4.1	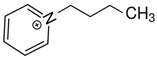 **BF_4_** 1-butyl-pyridinium	149.9 ± 2.3 [30]	−2.4	−23.2
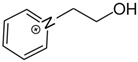 **OMs** 1-(2-hydroxyethyl)pyridinium	145.0 ± 2.1	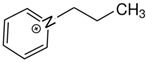 **OMs** 1-propyl-pyridinium	152.6 ± 2.1 ^c^	7.6	−32.7
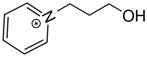 **OMs** 1-(3-hydroxypropyl)pyridinium	146.1 ± 2.1	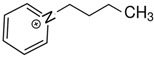 **OMs** 1-butyl-pyridinium	156.0 ± 2.1 ^c^	9.9	−35.5

^a^ Experimental data from Table 4. ^b^ Enthalpy of vaporisation of 1-propyl-pyridinium [DCA] is supposed to be equal to those of 3-methyl-1-propyl-pyridinium [DCA] derived in our previous work [31]. ^c^ Enthalpy of vaporisation was assessed as shown in Appendix A.

## Data Availability

All data are available in the text of manuscript and in Appendix A to this paper.

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
