# Peer review of "Molecular Liquids versus Ionic Liquids: The Interplay between Inter-Molecular and Intra-Molecular Hydrogen Bonding as Seen by Vaporisation Thermodynamics"

_molecules, 2023, doi:10.3390/molecules28020539_

Round 1

Reviewer 1 Report

Please find attached the review PDF file.

Reviewer 2 Report

The authors measured the enthalpies of vaporization of a large variety of molecular and ionic liquids including alkanols and alkanes, hydroxy-alkyl-piperidines and N-alkyl-piperidines, hydroxy-alkyl-benzene and alkyl-benzenes, and pyridinium based ionic liquids. The measured enthalpies of vaporisation were analysed by applying the well-known “group contribution” methods for molecular liquids and the “centerpiece” method for ionic liquids. This combined approach allowed to distinguish between inter- and intra-molecular hydrogen bonding of these liquids.

For example, for ionic liquids, the intra-HB strength is stronger than the inter-HB, indicating that the OH groups are mainly bound to the anions, and is somewhat dependent on the type of the anions.

This is an interesting paper and I enjoyed reading it. I think it can be published in this journal after miner revision.

1.      P. 6, L. 3: Should “intra” be “inter”?

2.      Table 4: only data for [N-C3H7-Py][NTf2] and [N-C4H9-Py][NTf2] were included, should data for [N-C2H7-Py][NTf2] also be included? Note that the data should agree with those in Table 5.

3.      P.12, L.25: Should “see Table 6, column 7 and 8” be “see Table 7, column 7 and 8”?

4.      Reference 30: the journal “ZAAC” should be listed before the published year 2017.
